# Isolation, Characterization, Genome Annotation, and Evaluation of Hyaluronidase Inhibitory Activity in Secondary Metabolites of *Brevibacillus* sp. JNUCC 41: A Comprehensive Analysis through Molecular Docking and Molecular Dynamics Simulation

**DOI:** 10.3390/ijms25094611

**Published:** 2024-04-23

**Authors:** Yang Xu, Xuhui Liang, Chang-Gu Hyun

**Affiliations:** Department of Beauty and Cosmetology, Jeju Inside Agency and Cosmetic Science Center, Jeju National University, Jeju 63243, Republic of Korea; iamxuyang1990@gmail.com (Y.X.); lxh03036@gmail.com (X.L.)

**Keywords:** *Brevibacillus*, isolation, genome annotation, hyaluronidase inhibitory activity, molecular docking, molecular dynamics simulation

## Abstract

*Brevibacillus* sp. JNUCC 41, characterized as a plant-growth-promoting rhizobacterium (PGPR), actively participates in lipid metabolism and biocontrol based on gene analysis. This study aimed to investigate the crucial secondary metabolites in biological metabolism; fermentation, extraction, and isolation were performed, revealing that methyl indole-3-acetate showed the best hyaluronidase (HAase) inhibitory activity (IC_50_: 343.9 μM). Molecular docking results further revealed that the compound forms hydrogen bonds with the residues Tyr-75 and Tyr-247 of HAase (binding energy: −6.4 kcal/mol). Molecular dynamics (MD) simulations demonstrated that the compound predominantly binds to HAase via hydrogen bonding (MM-PBSA binding energy: −24.9 kcal/mol) and exhibits good stability. The residues Tyr-247 and Tyr-202, pivotal for binding in docking, were also confirmed via MD simulations. This study suggests that methyl indole-3-acetate holds potential applications in anti-inflammatory and anti-aging treatments.

## 1. Introduction

*Brevibacillus* belongs to the family *Paenibacillaceae*. The genus *Brevibacillus* was originally proposed on the basis of the description of 10 species, including the type strain *Brevibacillus brevis* [1]. The genus *Brevibacillus* comprises Gram-positive or Gram-variable, oval, endospore-forming, rod-shaped bacteria with diverse ecological habitats, including antibiotic raw products, soil, sediment, microbiological agents, hot springs, and compost, as well as other habitats.

The genus *Brevibacillus* is considered a plant-growth-promoting rhizobacterium (PGPR) and has significant potential value in preventing plant diseases (in tea, tomatoes, cotton, corn, lettuce, and others), in soil bioremediation for heavy metal removal, etc. *Brevibacillus* has exhibited a broad spectrum of inhibitory effects in the field of biological control. *B. formosus* strain DSM 9885 and *B. brevis* strain NBRC 15304 demonstrated the potential to mitigate the impact of *Alternaria alternata*, thereby alleviating brown leaf spot disease in potatoes [2,3]. *B. laterosporus* AMCC100017 and *B. laterosporus* BL12 exhibited biocontrol capabilities against potato common scab [4,5]. *B. brevis* IPC11 exhibited inhibitory effects against bacterial canker disease in tomatoes, and *B. brevis* has been demonstrated as a potential biocontrol agent to reduce the impact of *Fusarium oxysporum* [6,7]. However, *Brevibacillus*, as a bioremediation factor, plays a crucial role in the removal of chemical materials and toxic metals and in reducing pollution in agricultural soil, water, and atmospheric environments [8]. A prominent example is the degradation of polyvinyl alcohol by *B. laterosporus* into acetate [9]. *B. laterosporus* can degrade plant tannins in wastewater from tanneries and biodegrade phenol and toluene [10,11]. *Brevibacillus* sp. may exert a significant role in safeguarding plants against Pb, Zn, and Ni toxicity via various mechanisms [12,13,14].

*Brevibacillus* spp. serve as a rich source of antimicrobial peptides (AMPs). Based on the biosynthetic pathways, AMPs can be categorized into ribosomally and nonribosomally synthesized groups [15]. The majority of *Brevibacillus* AMPs are nonribosomally synthesized, including Tostadin [16], Gramicidin A-C [17], Edeine [18], Spergualin [19], Tauramamide [20], Tridecapeptide family [21], Brevistin [22], and Brevibacillin [23]. Ribosomally synthesized *Brevibacillus* AMPs include Bac-GM100 [24], Laterosporulin [25], and Laterosporulin 10 [26]. A novel lanthipeptide of Brevicillin isolated from the genus *Brevibacillus* exhibits antimicrobial, antifungal, and antiviral activity [27]. The thiazoline derivatives Ulbactin F and G, isolated from *Brevibacillus* sp., demonstrate tumor cell migration inhibitory activity [28]. Ethylparaben, discovered from *Brevibacillus brevis* FJAT-0809-GLX, exhibits antimicrobial activity [29].

In microbiology, hyaluronic acid (HA) plays a crucial role in immune evasion, protecting bacteria from host and environmental factors [30,31]. Additionally, HA is thought to participate in physiological processes such as embryogenesis, cell migration, wound healing, tissue turnover, and malignancies [32,33,34]. Hyaluronidases are classes of glycosidases that primarily degrade HA. With the degradation of HA, exogenous inflammatory mediators infiltrate the body, causing an inflammatory response [35,36]. HAase inhibitors are involved in maintaining the balance between the anabolism and catabolism of HA, leading to the development of therapeutics of non-steroidal anti-inflammatory drugs (NSAIDs) [37]. The secondary metabolites associated with HAase inhibition include fatty acids, alkaloids, antioxidants, polyphenols, flavonoids, and terpenoids [38]. The human hyaluronidase 1 (HYAL 1, PDB ID: 2PE4) was studied using molecular docking methods to elucidate the mechanism regulating the enzyme’s function in the body [39].

The site at which flavonoids (myricetin, rutin, naringin, hesperidin, genistein, and puerarin) bind to HAase is near Tyr-75 (except for puerarin), Tyr-247, Tyr-286 (except for hesperidin), Tyr-202, Trp-321, and/or Trp-324, and these compounds form hydrogen bond interactions with Tyr-75, Tyr-247, and Tyr-286 [40]. Liquiritigenin forms two hydrogen bond interactions with the Tyr-286 and Asp-292 residues of HAase [41]. N-acetyl cysteine (NAC) forms hydrophobic interactions with Trp-130, Tyr-202, Phe-204, and Tyr-208. Glutathione (GSH) binds to the active site via hydrogen bonding with Gly-63, Tyr-247, Tyr-286, Trp321, and Trp-324 residues [42]. Glucoliquiritin apioside forms hydrogen bonds with Trp-324 and π–alkyl interactions with Tyr-202 [43]. There are three hydrogen bonds between baicalein and Asn-37, Tyr-75, and Trp-321. Similarly, three hydrogen bonds also exist between chrysin and Tyr-75, Tyr-286, and Trp-321 [44]. There are four hydrogen bonds between the –OH group of silybin and Asn-37, Val-127, Tyr-210, and Tyr-247 [45]. The synthesized eugenol 1,2,3-triazole derivatives exhibited strong binding interactions with Tyr-75 via hydrophobic interaction, Tyr-286 via π–π interaction, and other Pi-alkyl interactions with Tyr-247, Pro-62, and Ile-73 residues [46]. Synthesized betulinic acid derivatives form hydrogen bonding interactions with Arg-240, Arg-20, and Arg-195 [47].

In HAase, Glu-131 plays a crucial direct role in chemical catalysis and may be the proton donor for the hydroxyl-leaving group, while Asp-129 plays a supporting role [48]. Tyr-202 may bind to the substrate, and Tyr-247 is believed to coordinate and stabilize oxidation during transition state formation [49].

In this study, *Brevibacillus* sp. JNUCC 41 was isolated, characterized, and subjected to comprehensive gene functional annotation. To further investigate the secondary metabolites governing biological functions, the strain was fermented, extracted, and isolated. Compounds exhibiting hyaluronidase inhibitory activity were screened, followed by an in-depth investigation of the inhibitory mechanism using molecular docking and molecular dynamics (MD) simulations.

## 2. Results and Discussion

### 2.1. General Characteristics of the Genome of Brevibacillus sp. JNUCC 41

The genomic sequencing of *Brevibacillus* sp. JNUCC 41 was performed using an Illumina HiSeq sequencer, resulting in a total of 5,490,698 base pairs of sequences with a GC content of 40.33%. The complete genome of JNUCC 41 comprises 5185 CDSs, accounting for 92.21% of the total genes, along with 39 rRNA (13 5S, 13 16S, and 13 23S) and 84 tRNA operons. The assembled sequence has been submitted to and deposited in the NCBI GenBank database under the accession ID CP062163.

The illustration in Figure 1 provides a physical chromosome map indicating the positions of functional genes, the transcription occurring in both directions from specific sites, and the locations of replication origins (oriC). The genome atlas was drawn using CG View 1.0 (http://stothard.afns.ualberta.ca/cgview_server/, accessed on 21 November 2023) [50].

#### 2.1.1. COGs Database Annotations

The Clusters of Orthologous Genes (COGs) database has served as a widely utilized tool for microbial genome annotation and comparative genomics. A notable feature of the COG system is its classification, assigning all COGs to one of the 26 functional categories [51].

A total of 5185 protein-coding sequences were identified, of which 3478 were designated to COGs. Among the COGs categories in the strain JNUCC 41, seven had the largest proportions (each with ≥5% of the total COGs classifications): E (amino acid transport and metabolism, 513 open reading frames, ORFs, 9.89%), K (transcription, 408 ORFs, 7.87%), R (general function prediction only, 397 ORFs, 7.66%), G (carbohydrate transport and metabolism, 309 ORFs, 5.96%), J (translation, ribosomal structure, and biogenesis, 290 ORFs, 5.59%), I (lipid transport and metabolism, 274 ORFs, 5.28%), and H (coenzyme transport and metabolism, 273 ORFs, 5.27%). The detailed numbers of COGs functional categories are shown in Appendix A. In short, the majority of the proteins could be assigned to carbohydrate and amino acid transport and metabolism, reflecting the active metabolic ability of strain JNUCC 41 [52].

Moreover, 274 proteins were assigned to lipid transport and metabolism. Lipid transport and metabolism play crucial roles in bacterial metabolism, contributing to various cellular processes and functions (energy storage and source, membrane structure and fluidity, cell signaling, virulence and pathogenesis, environmental adaptation, etc.) [53,54]. Therefore, bacteria modulate their lipid composition in response to environmental changes, enabling them to adapt to different growth conditions and stresses [55].

#### 2.1.2. GO Database Annotations

Gene ontology (GO) analysis is a powerful bioinformatics tool that plays a significant role in elucidating the functional roles of genes and their products, including proteins, in bacterial metabolism [56].

The comparison of the JNUCC 41 strain’s genomic nucleotide sequence with GO database protein sequences reveals the presence of genes associated with three major types: cellular components, biological processes, and molecular functions. A total of 2999 genes have been annotated in the GO database, accounting for 57.8% of the total coding genes. GO analysis suggested that the biological-process-related genes (1561 ORFs, 30.11%) were the most abundant in strain JNUCC 41, followed by molecular function (605 ORFs, 11.17%) and cellular component (833 ORFs, 17.03%).

Among the sub-functions annotated by GO analysis, the metabolic process (521 ORFs, 10.5%) and cellular process (594 ORFs, 11.46%) were dominant in the biological process category. Catalytic activity (448 ORFs, 8.64%) and binding (187 ORFs, 3.61%) were the core functions in the molecular function category, and cellular anatomical entity (518 ORFs, 9.99%) was dominant in the cellular component category, as shown in Figure 2.

In summary, the above results provide a deeper understanding of the molecular basis of bacterial physiology, adaptation to different environments, and interactions with hosts or ecosystems.

#### 2.1.3. KEGG Database Annotations

The KEGG database was developed as a reference knowledge base to reveal cellular and organismal functions from genome sequences and other molecular datasets [57]. The strain JNUCC 41 has 2267 genes annotated in the KEGG database, including six primary functional categories: metabolism, genetic information processing, environmental information processing, cellular processes, organismal systems, and human diseases (Figure 3).

According to the KEGG database, the genome of strain JNUCC 41 contained a variety of functional genes related to metabolism (1454 ORFs, 28.14%), including amino acid metabolism (317 ORFs, 6.11%), carbohydrate metabolism (276 ORFs, 5.32%), the metabolism of cofactors and vitamins (235 ORFs, 4.53%), energy metabolism (162 ORFs, 3.12%), lipid metabolism (96 ORFs, 1.85%) and nucleotide metabolism (96 ORFs, 1.85%). A total of 308 ORFs have been annotated in the environmental information processing category, with 178 ORFs related to membrane transport and 129 ORFs associated with signal transduction.

The extensive annotation of metabolism in the strain JNUCC 41 demonstrates its metabolic capabilities and offers insights into the biochemical processes within bacterial cells, encompassing the synthesis and breakdown of various molecules, energy production, and other metabolic activities.

#### 2.1.4. CAZy Database Annotations

A total of 116 proteins were annotated as CAZymes using the dbCAN meta-server [58,59]. As shown in Appendix A, among the five functional family domains in strain JNUCC 41, glycosyl transferases and glycoside hydrolases were the most abundant (32 domains each), followed by carbohydrate-binding modules (26 domains), carbohydrate esterases (17 domains), and auxiliary activity (9 domains). The genomic analysis revealed the presence of various glycoside hydrolase families (GH1, GH3, GH4, GH8, GH13, GH15, GH18, GH23, GH25, GH73, GH109, GH170, GH171, and GH179) encoding cell-wall-degrading enzymes.

The study suggests that glycoside hydrolases, including GH13, GH2, and GH43, assist Vibrio parahaemolyticus in degrading algal extracellular polysaccharides. Additionally, certain bacteria within the Firmicutes phylum utilize enzymes such as GH1 for polysaccharide degradation [60,61]. Beneficial bacteria utilize key carbohydrate metabolism genes to metabolize algal polysaccharides, facilitating material exchange with algae and potentially aiding in algae’s resistance to external stress via substance secretion or other mechanisms, thereby achieving biological control [62].

Previous studies have indicated that glycoside hydrolase activity can reduce the biomass of adherent biofilms, resulting in the secretion of low-molecular-weight Pel extracellular polysaccharides. This activity has the potential to decrease the virulence of *Pseudomonas aeruginosa* in Caenorhabditis elegans and Drosophila melanogaster. The glycoside hydrolase family PeBgl1 has been shown to reduce the pathogenicity of *Penicillium expansum* on apples [63]. In summary, it is tentatively speculated that the strain JNUCC 41 inhibits the proliferation of pathogenic bacteria by disrupting the integrity of the pathogen’s cell wall.

#### 2.1.5. Virulence Factor Gene Mining

A total of 43 putative virulence genes associated with the adhesion, colonization, and destruction of tissues were found in the strain JNUCC 41 using the virulence factor database (VFDB) [64]. As shown in Appendix A, the VFDB functional categories revealed that the most abundant families were immune modulation (23 domains), followed by nutritional/metabolic factor (5 domains), adherence (4 domains), regulation (3 domains), exoenzyme (2 domains), exotoxin (2 domains), effector delivery system (2 domains), and motility (1 domain).

Immune modulation involves hyaluronic acid, polyglutamic acid, and polysaccharide capsules, with previous studies showing bacteria like *Streptococcus* and *Pasteurella multocida* produce hyaluronic acid in their capsules and mucus [65]. In vivo, *Bacillus anthracis* secretes a polypeptide capsule called polyglutamic acid, which protects it from phagocytosis and partially resists the bactericidal effects of human defensins [66]. Polysaccharide capsules have the potential to mediate various biological processes, including invasive infections in humans [67]. Additionally, fibronectin-binding protein (FbpA) is involved in adherence, and studies have shown its effective antibacterial activity against *Staphylococcus aureus* infections in the mammary gland [68,69].

### 2.2. Secondary Metabolites Isolated from Brevibacillus sp. JNUCC 41

A total of five known compounds were obtained from the purification of the ethyl acetate extract (600 mg) of the culture broth of strain JNUCC 41, including methyl indole-3-acetate (**1**) [70], dibutyl phthalate (**2**) [71], daidzein (**3**) [72], maculosin (**4**) [73], and N-Acetyl-L-tryptophan (**5**) [74], as shown in Figure 4.

### 2.3. Hyaluronidase Inhibitory Activity of the Isolated Secondary Metabolites

The hyaluronidase inhibitory activities of all compounds were screened. Among them, methyl indole-3-acetate showed hyaluronidase inhibitory activity with an IC_50_ value of 343.9 μM. EGCG were used as positive controls, with IC_50_ values of 172.3 μM. (Table 1, Figure 5).

### 2.4. Molecular Properties and Drug Likeness

The ADMET properties and drug-likeness properties of the compounds were obtained via a web server, as shown in Appendix A. The physicochemical properties indicate the high CaCO-2 permeability of methyl indole-3-acetate. It does not act as an inhibitor or substrate of P-glycoprotein (p-gp). It only inhibits CYP1A2, whereas EGCG is a CYP3A4 inhibitor. The topological polar surface area (TPSA) is correlated with the transmembrane transport characteristics, blood–brain barrier penetration, and intestinal permeability prediction [75]. Molecules with a TPSA/PSA of ≤160 Å^2^ demonstrate good intestinal absorption, while those with a TPSA/PSA ≤ 60 Å^2^ exhibit blood–brain barrier (BBB) permeability [76]. The TPSA of methyl indole-3-acetate is less than 160 Å^2^, and the human intestinal absorption rates are more than 90%, far surpassing EGCG. Additionally, methyl indole-3-acetate displays TPSA ≤ 60 Å^2^, suggesting that it is a better BBB agent.

The predicted plasma protein binding (PPB) of 56.81% suggests moderate affinity of the molecule for plasma transport proteins. This may elevate levels of the active form in tissues, enhancing pharmacological effects by facilitating improved tissue penetration and interaction with target sites. Additionally, moderate PPB could minimize interference in pharmacokinetic processes.

The predicted half-life of methyl indole-3-acetate is 0.898 h, and the clearance rate is 10.460 mL/min/kg. Regarding toxicity, methyl indole-3-acetate did not reveal hepatotoxicity, skin sensitization, Ames toxicity, or hERG inhibition, while EGCG exhibited hERG inhibition. The lethal toxicity dose of all compounds was high. In summary, the results indicate that the compound methyl indole-3-acetate has certain advantages in ADMET properties compared with the reference compound.

With a molecular weight of 189.08 g/mol, a log *p* of 2.2, one hydrogen-bond donor, and three hydrogen-bond acceptors, methyl indole-3-acetate showed zero violations of Lipinski’s rule [77]. The Ghose Filter values were −0.4 ≤ log *p* ≤ 130, 40 ≤ molar refractivity ≤ 130, 160 ≤ molecular weight ≤ 480, 20 ≤ number of atoms ≤ 70, and TPSA ≤ 140 Å^2^ [78], the Veber Filter considers good bioavailability for compounds with TPSA ≤ 140 Å^2^ and number of rotatable bonds ≤ 10 [79], and the Egan Filter considers drug candidates to have good oral bioavailability when the values are −1.0 ≤ log *p* ≤ 5.8 and TPSA ≤ 130 Å^2^ [80]. The results show that methyl indole-3-acetate complies with the Ghose Filter, Veber Filter, Egan Filter, and Lipinski’s rule. Overall, methyl indole-3-acetate has better drug-likeness properties compared to the reference compound.

### 2.5. Docking and Molecular Dynamics (MD) Simulations

#### 2.5.1. Molecular Docking

Generally, if the change in system free energy is negative, protein–ligand binding can occur spontaneously. A binding energy of more negative than −5 kcal/mol signifies excellent binding affinity. To analyze the inhibition mechanism, molecular docking was conducted utilizing the internal cavity of HAase (PDB ID: 2PE4) as the active site [81]. In this docking study of HYAL 1 with methyl indole-3-acetate, the binding energy was calculated to be −6.4 kcal/mol, indicating a high degree of binding between HYAL 1 and methyl indole-3-acetate. The visualization of the docking results using PyMOL 2.3.0 and Discovery Studio 2019 [82,83] is shown in Figure 6. The results reveal that methyl indole-3-acetate interacts with the amino acid residues Tyr-75 and Tyr-247 in chain A of HYAL 1, forming two hydrogen bonds (represented by green dashed lines) with bond distances of 2.35Å and 2.95Å, respectively. A hydrogen bond distance below 3.0 Å indicates stronger mutual attraction between the hydrogen atom and the electronegative atom, hence implying stronger interaction forces. Additionally, there is a π–π stacking interaction with Tyr-202 (purple dashed line) and two π–alkyl interactions with Ile-73 and Tyr-202 (pale pink dashed lines).

The docking results reveal several key interactions: Firstly, the carbonyl group of methyl indole-3-acetate forms hydrogen bonds with Tyr-247, a residue recognized for stabilizing oxidation during transition state formation [49], and the amino group interacts with Tyr-75 via hydrogen bonding. Secondly, the phenyl group interacts with Tyr-202, a hydrophobic amino acid likely involved in substrate binding [84], via π–π stacking and π–alkyl interactions. Thirdly, interactions occur between the compound and hydrophobic amino acid residues, specifically Asp-129 and Glu-131, recognized for their chemical catalysis role [48], forming π–anion and carbon–hydrogen interactions, respectively. The importance of Glu-131, Asp-129, Tyr-202, and Tyr-247 is underscored by mutagenesis studies, which demonstrate impaired enzyme activity upon their alteration. These findings suggest that methyl indole-3-acetate occupies the catalytic active site of HAase, potentially leading to decreased catalytic activity.

The results indicate that in the ten repeated docking experiments, as shown in Figure 7, methyl indole-3-acetate consistently binds to HYAL 1 at the same location with minimal conformational differences. Furthermore, the fluctuation in docking binding energies within ±0.1 kcal/mol suggests high reproducibility, affirming the feasibility of the molecular docking method.

However, the semi-flexible docking approach used in molecular docking cannot account for the flexibility of protein structures. To further validate the degree and stability of binding between ligands and proteins, this study conducted a 100 ns molecular dynamics simulation of the complex between HYAL 1 and methyl indole-3-acetate.

#### 2.5.2. Molecular Dynamics (MD) Simulations

The RMSD is a critical parameter for assessing the stability of protein–ligand complexes, where smaller RMSD values indicate minimal overall structural changes in the complex, thus indicating better stability of the complex [85]. As shown in Figure 8a, the RMSD curve of the complex between HYAL 1 and methyl indole-3-acetate fluctuates within a range of 1 nm during the process of MD simulations, with no significant fluctuations observed. The RMSD analysis suggests that the structural flexibility of HYAL 1’s active site regions is crucial in facilitating conformational changes induced by ligand binding, particularly evident in the complex formed with methyl indole-3-acetate.

The RMSF represents the extent of the fluctuation of amino acid residues in the protein during MD simulations [86]. Higher RMSF values indicate greater fluctuations in the amino acid residues, whereas lower RMSF values indicate smaller fluctuations. As shown in Figure 8b, the RMSF curves of the complex of HYAL 1 with methyl indole-3-acetate fluctuate within a range of 1 nm, with no significant fluctuations observed. Only the terminal amino acids exhibit reasonable fluctuations close to 0.5 nm. This indicates that the addition of methyl indole-3-acetate has minimal impact on the stability of amino acid residues in the HYAL 1, and the stability of the formed complex is excellent.

Rg is used to characterize the compactness and stability of structures, with a larger Rg indicating a more severe expansion of the system during MD simulations, whereas a smaller Rg suggests that the system remains compact and stable during MD simulations [87]. As shown in Figure 9a, the Rg curve of the complex formed between HYAL 1 and methyl indole-3-acetate consistently fluctuates within the range of 2.2–2.3 nm throughout the simulation, maintaining a compact conformation without notable deviations. This observation suggests that the interaction between methyl indole-3-acetate and HYAL 1 leads to a compact binding without inducing significant structural changes in the overall protein conformation upon methyl indole-3-acetate binding.

To investigate the hydrogen-bonding properties of the binding site of the complex, this study calculated the number of hydrogen bonds between the ligand and the protein, which play a major role in stabilizing the complex [88]. As shown in Figure 9b, after 20 ns, the number of hydrogen bonds between HYAL 1 and methyl indole-3-acetate stabilized at 1-2, with smooth fluctuations in the hydrogen bond curve, indicating a good hydrogen bond interaction between HYAL 1 and methyl indole-3-acetate.

The SASA is one of the factors used to study protein structure folding and stability [89]. Proteins with stable structures typically exhibit more stable SASA curves. As shown in Figure 10a, the SASA curve of the complex between HYAL 1 and methyl indole-3-acetate remains stable throughout the simulation without significant fluctuations, indicating the high stability of the complex.

The Gibbs FEL was calculated using the built-in Gromacs scripts g_sham and xpm2txt.py. The Gibbs relative free energy was obtained based on RMSD and Rg values and plotted as X, Y, and Z axes to generate the Gibbs FEL [90]. The Gibbs FEL is used to describe the energetically favored conformations throughout the entire MD simulation of the complex. Weak or unstable interactions between the protein and ligand result in multiple and rough energy clusters on the Gibbs FEL, whereas strong and stable interactions form nearly single and smooth energy clusters. In Figure 10b, dark purple/blue spots reflect stable structures with the lowest energy values, while red/yellow spots represent unstable structures. The results demonstrate that the Gibbs FEL of the complex between HYAL 1 and methyl indole-3-acetate forms a single and concentrated energy cluster, indicating the good stability of the complex.

After achieving stability in the complex system, we computed the MM/PBSA-binding energy [91] of the HYAL 1 with methyl indole-3-acetate complex. In the gas phase, interactions primarily consist of van der Waals forces and electrostatic interactions. In MM-PBSA calculations, gas phase Gibbs free energy (GGAS) can be computed as the sum of van der Waals energy (VDWAALS) and electrostatic energy (EEL). with VDWAALS at −24.15 kcal/mol and EEL at −25.01 kcal/mol, GGAS is calculated as −49.17 kcal/mol (Figure 11a). Typically, solvent effects are considered via solvent polarization energy (EGB) and surface tension energy (ESURF), which contribute to solvent Gibbs free energy (GSOLV). With EGB at 27.83 kcal/mol and ESURF at −3.56 kcal/mol, GSOLV is calculated as 24.27 kcal/mol. The total free energy (ΔG) in MM-PBSA calculations is the sum of GSOLV and GGAS. Negative ΔG values denote stable binding, indicating strong ligand-receptor interactions, while positive values suggest instability or repulsion. A final MM-PBSA result of −24.9 kcal/mol indicates a stronger binding affinity.

The results of the residue–energy analysis [92] demonstrated that the methyl indole-3-acetate ligand binds most effectively to the amino acid residues Tyr-202, Tyr-247, Tyr-286, and Trp-321 in the HYAL 1, indicating that Tyr-202, Tyr-247, Tyr-286, and Trp-321 play a major role in the interaction between the methyl indole-3-acetate ligand and HYAL 1. The results are shown in Figure 11b.

To further analyze the binding status of the complexes during the MD simulation process, the complex structures at 25 ns, 50 ns, 75 ns, and 100 ns in the MD simulation trajectory of the methyl indole-3-acetate were extracted and compared. Throughout the MD simulation process, it was evident that methyl indole-3-acetate consistently occupies the internal cavity of the active binding site in the HYAL 1 without significant changes, indicating that the formed complex exhibits good stability, as depicted in Figure 12.

## 3. Materials and Methods

### 3.1. Isolation of the Bacterial Brevibacillus sp. JNUCC 41

*Brevibacillus* sp. JNUCC 41 was isolated from Baengnokdam, the summit crater of Mount Halla on Jeju Island, Republic of Korea. The initial identification of the strain was performed using 16S ribosomal RNA gene sequencing. Soil samples (0.5 g) were suspended in 0.45 mL of 0.1% Tris buffer (*w*/*v*) and agitated at 180 rpm at 30 °C for 1 h. Subsequently, 100 µL of the suspension was subjected to serial dilution (10^−5^ to 10^−9^) and plated onto MRS medium. For routine culture, the strain JNUCC 41 was cultured aerobically on LB solid medium and LB liquid broth for 1 day at 30 °C, with preservation in a 20% (*v*/*v*) glycerol suspension at −80 °C.

### 3.2. Genome Extraction and Sequencing

The QIAGEN genomic-tip Kit (Qiagen Inc., Shenzhen, China) was used to extract whole-genomic DNA from solid colonies of strain JNUCC 41. The genome was sequenced using PacBio RSII and an Illumina platform at Macrogen, Inc. (Seoul, Republic of Korea). The presence of plasmids in the genome was assessed using PlasmidFinder 2.1.

### 3.3. Genome Annotation

The genes obtained were subjected to comparison with COGs (https://www.ncbi.nlm.nih.gov/COG/, accessed on 21 November 2023), GO (https://geneontology.org/, accessed on 21 November 2023), and KEGG (https://www.kegg.jp/, accessed on 21 November 2023) databases for functional annotation information using BLAST, BLAST2GO, and Diamond sequence alignment tools. The parameter E-value for Diamond was set to 10^−5^. To predict virulence genes, antibiotic resistance genes, and carbohydrate-active enzymes in strain JNUCC 41, the Virulence Factor Database (VFDB) (http://www.mgc.ac.cn/cgi-bin/VFs/v5/main.cgi, accessed on 21 November 2023) was used for virulence gene detection, and the Carbohydrate-Active Enzymes database (CAZy) (https://bcb.unl.edu/dbCAN2/blast.php/, accessed on 21 November 2023) was used for carbohydrate-active enzyme detection. http://www.mgc.ac.cn/VFs/main.htm, accessed on 21 November 2023.

### 3.4. General Experimental Procedures

Luria–Bertani Broth (LB) and Lactobacilli MRS Broth (Becton, Franklin Lakes, NJ, USA) were used for the bacterial culture. Silica gel (Merck, Darmstadt, Germany) and Sephadex LH-20 gel (Sigma-Aldrich, St. Louis, MO, USA) were used for chromatographic separations. Thin-layer chromatography (TLC) plates (Merck, Darmstadt, Germany) were used for analytical purposes. Hyaluronidase was obtained from bovine testes (Sigma-Aldrich, St. Louis, MO, USA). Nuclear magnetic resonance (NMR) spectrums were obtained using JNM-ECX 400 (FT-NMR system, 400 MHz, JEOL Co., Akishima, Japan).

### 3.5. Fermentation, Extraction, and Isolation

The strain JNUCC 41 was cultured in a 250 mL flask containing 125 mL of LB medium at 30 °C for 48 h in a shaking incubator. Subsequently, the culture was scaled up to four 5 L flasks, each containing 1 L of LB medium, with an initial inoculum volume of 5% (*v*/*v*). The culture solution (4 L) of strain JNUCC 41 underwent extraction with EtOAc (4 L × 3 times). The EtOAc extraction was concentrated under reduced pressure to yield a residue (600 mg). Subsequently, The EtOAc soluble fraction was subjected to vacuum liquid chromatography (VLC) on silica gel using a step gradient (CHCl_3_–MeOH, 300 mL each) to provide 10 fractions (Fr. V1–V10).

Compound **1** (10.4 mg), Compound **2** (11.2 mg), and Compound **3** (13.8 mg) were isolated from combined Fr. V5 using silica gel column chromatography (CC) with CHCl_3_–MeOH (50:1, *v*/*v*). Compounds **4** (16.2 mg) and **5** (10.7 mg) were isolated from combined Fr. V9 using silica gel column chromatography (CC) with CHCl_3_–MeOH (20:1, *v*/*v*). The nuclear magnetic resonance (NMR) data of the isolated compounds are presented in Appendix A.

#### 3.5.1. Methyl Indole-3-Acetate

^1^H NMR (400 MHz, CHLOROFORM-*D*) δ 7.62 (ddt, *J* = 7.8, 1.5, 0.8 Hz, 1H, H-4), 7.34 (dt, *J* = 8.1, 1.0 Hz, 1H, H-7), 7.21 (ddd, *J* = 8.1, 7.0, 1.3 Hz, 1H, H-1), 7.17–7.12 (m, 2H, H-5 and H-6), 3.79 (d, *J* = 0.9 Hz, 2H, H-9), 3.71 (s, 3H, H-11). ^13^C NMR (101 MHz, CHLOROFORM-*D*) δ 172.71 (C-10), 136.19 (C-8), 127.31 (C-3), 123.20 (C-1), 122.33 (C-6), 119.81 (C-5), 118.95 (C-4), 111.33 (C-7), 108.51 (C-2), 52.12 (C-11), and 31.26 (C-9).

#### 3.5.2. Dibutyl Phthalate

^1^H NMR (400 MHz, CHLOROFORM-*D*) δ 7.71 (dd, *J* = 5.7, 3.3 Hz, 2H, H-3 and H-4), 7.57 –7.48 (m, 2H, H-2 and H-5), 4.30 (t, *J* = 6.7 Hz, 4H, CH_2_-2′ and CH_2_-2″), 1.77–1.66 (m, 4H, CH_2_-3′ and CH_2_-3″), 1.51–1.37 (m, 4H, CH_2_-4′ and CH_2_-4″), 0.96 (t, *J* = 7.4 Hz, 6H, CH_3_-5′ and CH_3_-5″). ^13^C NMR (101 MHz, CHLOROFORM-*D*) δ 167.55 (C-1′ and C-1″), 132.11 (C-1 and C-6), 130.74 (C-3 and C-4), 128.65 (C-2 and C-5), 65.38 (C-2′ and C-2″), 30.37 (C-3′ and C-3″), 18.99 (C-4′ and C-4″), and 13.55 (C-5′ and C-5″).

#### 3.5.3. Daidzein

^1^H NMR (400 MHz, DMSO-*D_6_*) δ 10.80 (s, 1H, H-5′), 9.55 (s, 1H, H-11), 8.29 (s, 1H, H-2), 7.97 (d, *J* = 8.8 Hz, 1H, H-5), 7.42–7.36 (m, 2H, H-2′ and H-2″), 6.94 (dd, *J* = 8.8, 2.3 Hz, 1H, H-6), 6.86 (d, *J* = 2.2 Hz, 1H, H-8), 6.84–6.78 (m, 2H, H-3′ and H-3″). ^13^C NMR (101 MHz, DMSO-*D_6_*) δ 174.74 (C-4), 162.55 (C-7), 157.47 (C-9), 157.22 (C-4′), 152.87 (C-2), 130.13 (C-2′ and C-2″), 127.34 (C-5), 123.52 (C-3), 122.58 (C-1′), 116.67 (C-10), 115.16 (C-3′ and C-3″), 114.99 (C-6), and 102.14 (C-8).

#### 3.5.4. Maculosin

^1^H NMR (400 MHz, METHANOL-*D_3_*) δ 7.10–6.98 (m, 2H, H-10 and H-14), 6.75–6.66 (m, 2H, H-11 and H-13), 4.36 (td, *J* = 4.9, 1.9 Hz, 1H, H-6), 4.04 (ddd, *J* = 11.0, 6.3, 1.9 Hz, 1H, H-3), 3.54 (dt, *J* = 11.9, 8.3 Hz, 1H, H-4a), 3.36 (d, *J* = 5.9 Hz, 1H, H-4b), 3.13–2.97 (m, 2H, H-8), 2.16–2.00 (m, 1H, H-2a), 1.80 (tdd, *J* = 8.4, 6.6, 4.6 Hz, 2H, H-1), 1.29–1.15 (m, 1H, 2b). ^13^C NMR (101 MHz, METHANOL-*D_3_*) δ 170.78 (C-5), 166.95 (C-7), 157.66 (C-12), 132.12(C-10 and C-14), 127.62 (C-9), 116.17 (C-11 and C-13), 60.04 (C-3), 57.88 (C-6), 45.91 (C-4), 37.63 (C-8), 29.38 (C-2), and 22.71 (C-1).

#### 3.5.5. N-Acetyl-L-Tryptophan

^1^H NMR (400 MHz, DMSO-*D_6_*) δ 10.84 (d, *J* = 2.3 Hz, 1H, H-1′), 8.17 (d, *J* = 7.8 Hz, 1H, H- 4), 7.53 (dd, *J* = 7.8, 1.1 Hz, 1H, H-6′), 7.33 (dt, *J* = 8.0, 0.9 Hz, 1H, H-7′), 7.14 (d, *J* = 2.4 Hz, 1H, H-5′), 7.06 (ddd, *J* = 8.1, 7.0, 1.2 Hz, 1H, H-2′), 6.98 (ddd, *J* = 7.9, 6.9, 1.1 Hz, 1H, H-4′), 4.51–4.41 (m, 1H, H-2), 3.16 (dd, *J* = 14.6, 5.0 Hz, 1H, H-1), 2.98 (dd, *J* = 14.6, 8.8 Hz, 1H, H-1), 1.80 (s, 3H, H-6). ^13^C NMR (101 MHz, DMSO-*D_6_*) δ 173.65(C-3), 169.31(C-5), 136.14(C-8′), 127.24 (C-9′), 123.59 (C-5′), 120.97 (C-4′), 118.42 (C-2’), 118.21 (C-6′), 111.44 (C-7′), 110.02 (C-3′), 53.03 (C-2), 27.18(C-6), and 22.46 (C-1).

### 3.6. Hyaluronidase Inhibitory Activity of Secondary Metabolites

In the enzymatic assay, 5 μL of hyaluronidase enzyme and 12.5 mM CaCl_2_ samples at varying concentrations were added to a 96-well plate. The samples were reacted for 20 min and incubated at 37 °C. After the initial incubation, 25 μL of the substrate solution (HA at a concentration of 2.4 mg/mL) was added. The reaction mixture was incubated again at 37 °C for 40 min. Then, 5 μL of 0.4 N NaOH and 5 μL of 0.4 M potassium tetraborate were added to the mixture, followed by incubation at 100 °C for 3 min. After cooling, 150 μL of DMAB solution (*p*-dimethylaminobenzaldehyde + acetic acid + 10 N HCL) was added to the reaction mixture. The absorbance of the reaction mixture was measured at 540 nm. Epigallocatechin gallate (EGCG) was used as a positive control in this assay.

### 3.7. Molecular Properties and Drug Likeness

Pharmacokinetic parameters were assessed via the analysis of ADMET and drug-likeness properties. By submitting the SMILES information of the compound to a prediction website, obtain the ADMET and drug-likeness properties of the compound. The ADMET properties of the compounds were comprehensively investigated utilizing various models, including ADMETlab 2.0, admetSAR 2.0, and pkCSM. The drug-likeness properties of the compounds were assessed using SwissADME (http://www.swissadme.ch/, accessed on 18 February 2024).

### 3.8. Molecular Docking Simulation

The 3D structure of the receptor protein human hyaluronidase 1 (PDB ID: 2PE4) was obtained from the Protein Data Bank (PDB) (http://www.rcsb.org/, accessed on 29 January 2024). PyMOL 2.3.0 software was used to inspect the protein structure for docking purposes. The 3D structure of the ligand methyl indole-3-acetate (PubChem CID: 74706) was downloaded from the PubChem database (https://pubchem.ncbi.nlm.nih.gov/, accessed on 29 January 2024), and the molecular structure was optimized using the MMFF94 force field in OpenBabel 2.4.1 software to obtain the optimal molecular structure in its lowest energy state [93].

Using AutoDock Tools 1.5.6 [94], the protein was subjected to hydrogenation treatment, and the ligand molecules were hydrogenated and treated to identify rotatable bonds and saved as pdbqt files. The docking parameters were set as follows: center_x = 41.8, center_y = −19.2, center_z = −16.2, size_x = 28, size_y = 28, and size_z = 28. The docking mode was configured for semi-flexible docking, with an exhaustiveness of 25, and the Lamarckian Genetic Algorithm (LGA) as the docking algorithm. Molecular docking was performed using AutoDock Vina 1.2.0 software to obtain the binding free energy and docking result files. To validate the reliability of molecular docking, this study conducted ten repeated docking simulations of the protein–ligand complex under identical conditions and compared the results.

### 3.9. Molecular Dynamics (MD) Simulations

The complex of HYAL 1 and methyl indole-3-acetate was subjected to molecular dynamics simulations using Gromacs 2022 software. The Amber14sb force field was applied to the protein, while the Gaff2 force field was used for the ligand. The SPC/E water model was employed to solvate the protein–ligand system, and a periodic boundary box with a size of 1.2 nm was established. The particle mesh Ewald (PME) method was utilized to compute long-range electrostatic interactions. To neutralize the system’s charge, an appropriate number of sodium and chloride ions were introduced using the Monte Carlo ion placement method.

Prior to the formal simulation, the system underwent energy minimization and equilibration via the following three steps: (1) The energy minimization of each system was performed using the steepest descent algorithm with 50,000 steps (minimization stopped when the maximum force < 1000 kJ/mol). (2) Each system underwent a 50,000-step NVT ensemble simulation, maintaining the number of particles, temperature at 310 K, and a time step of 2 fs. (3) A 50,000-step NPT ensemble simulation was conducted for each system, with the temperature set at 310 K, pressure at 1 atm, and a time step of 2 fs. The simulation was then continued for 100 ns, whereby the coordinates were saved every 10 ps for analysis.

Ultimately, we analyzed the MD simulation trajectory of the HYAL 1–methyl indole-3-acetate complex, focusing on the root mean square deviation (RMSD), root mean square fluctuation (RMSF), solvent-accessible surface area (SASA), radius of gyration (Rg), hydrogen-bond analysis (H-bond), Gibbs free energy landscape (FEL), molecular mechanics–Poisson–Boltzmann surface area (MM-PBSA) binding energy, and energy contributions from amino acids involved in binding (residue–energy). Additionally, we conducted structural comparisons of the complex at time points of 25, 50, 75, and 100 ns.

## 4. Conclusions

The genome functional annotation of *Brevibacillus* sp. JNUCC 41 indicates robust metabolic capabilities, thus prompting the exploration of secondary metabolites that play significant roles in biological metabolism. Among the secondary metabolites, methyl indole-3-acetate exhibited better HAase inhibitory activity and demonstrated favorable ADMET and drug-likeness properties compared to the reference compound EGCG.

The results of molecular docking simulations on the inhibition mechanism showed that methyl indole-3-acetate forms two hydrogen bonds with amino acid residues Tyr-247 and Tyr-75 and interacts with various hydrophobic amino acid residues within the active site, thereby inhibiting the activity of HYAL 1. MD simulation revealed that the complex formed between HYAL 1 and methyl indole-3-acetate exhibits remarkable stability, as evidenced by RMSD, RMSF, Rg, SASA, and Gibbs FEL. Hydrogen bond analysis further supports the stability of the complex through favorable interactions. The MM/PBSA binding energy calculation (−24.9 kcal/mol) underscores a robust binding affinity. The residue–energy analysis identified Tyr-247 and Tyr-202 as crucial residues mediating the binding of methyl indole-3-acetate to the HYAL 1, consistent with molecular docking predictions highlighting their pivotal roles in the interaction. In general, methyl indole-3-acetate, as an active molecule inhibiting HAase, exhibits promising potential for applications in anti-inflammatories and cosmetics (anti-aging, wrinkle reduction, and moisturizers).

## Figures and Tables

**Figure 1 ijms-25-04611-f001:**
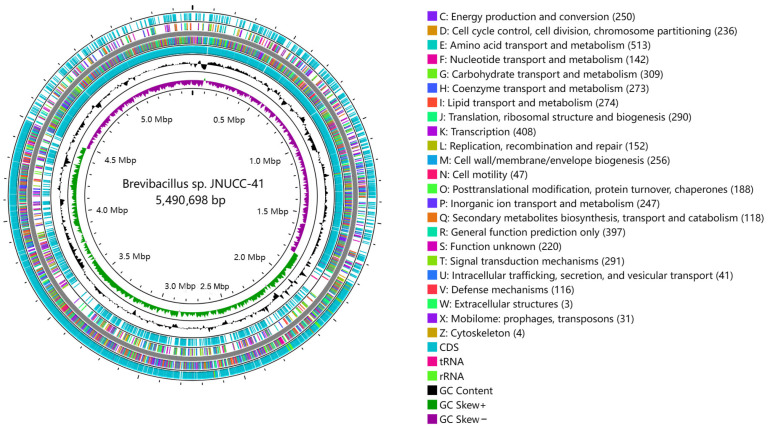
Circular map of the *Brevibacillus* sp. JNUCC 41 strain chromosome. Summary of gene annotation and GC skew analysis of the genome of the strain JNUCC 41. Circles (from inner to outer): circle 1 represents the scale; circle 2 shows the GC skew; circle 3 shows the GC content; circles 4 and 7 show CDS, rRNA, and tRNA on the positive and negative strands; and circles 5 and 6 show the COGs of each of the coding sequences (CDSs).

**Figure 2 ijms-25-04611-f002:**
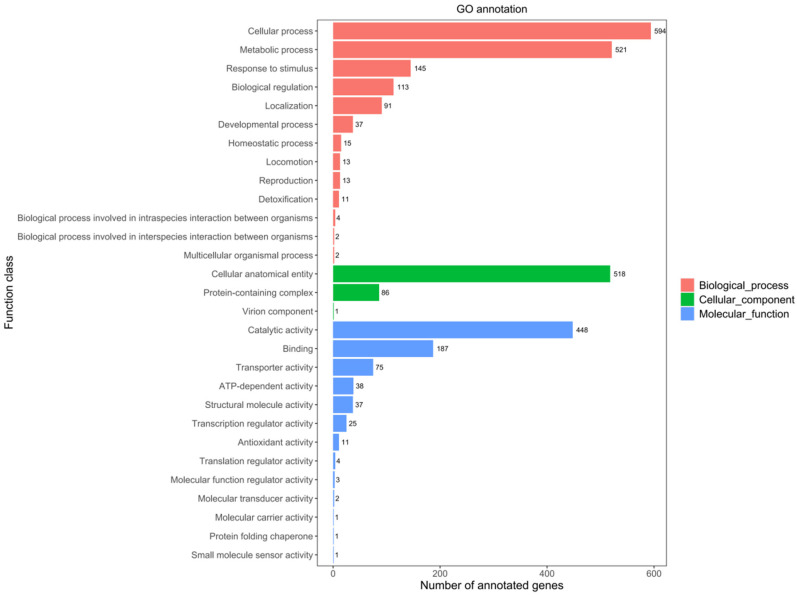
GO functional classification of the genome.

**Figure 3 ijms-25-04611-f003:**
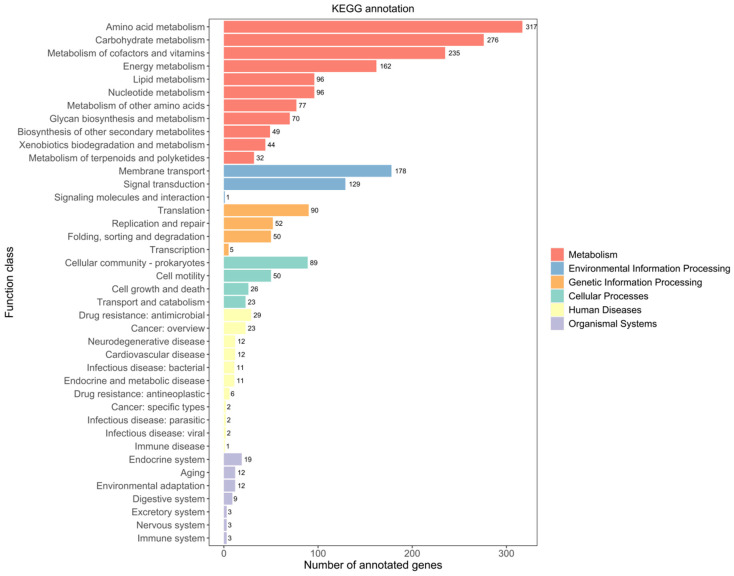
KEGG functional classification of the genome.

**Figure 4 ijms-25-04611-f004:**
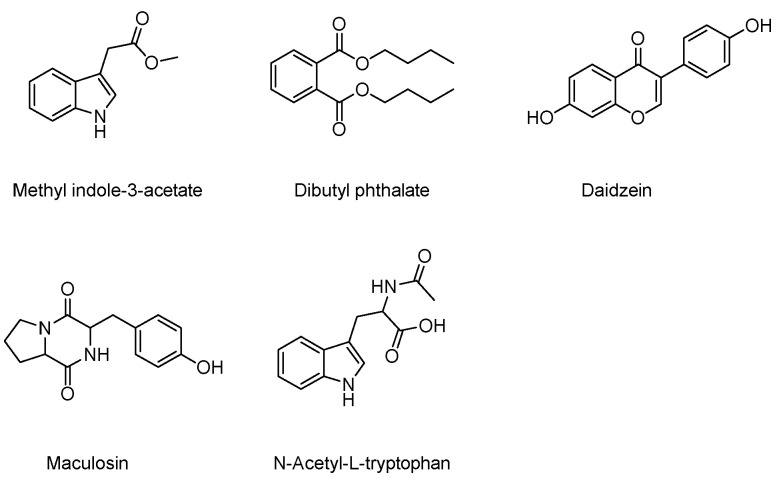
Chemical structure of the isolated compounds from the strain JNUCC 41.

**Figure 5 ijms-25-04611-f005:**
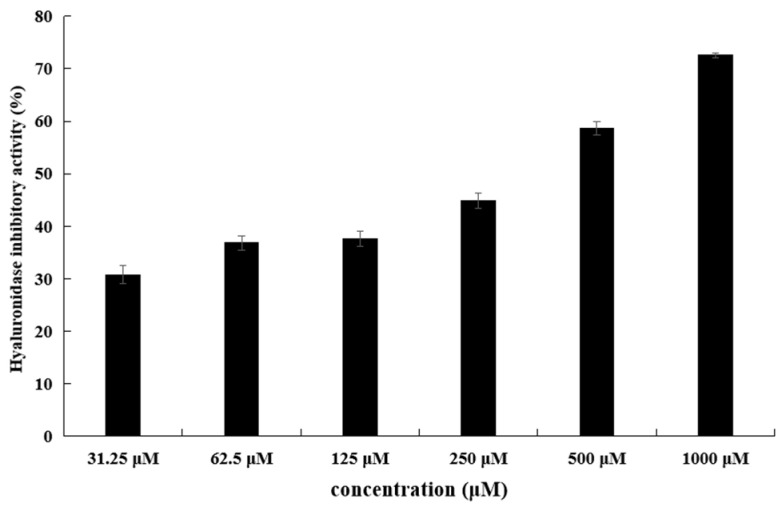
Hyaluronidase-inhibitory activity of methyl indole-3-acetate from the strain JNUCC 41.

**Figure 6 ijms-25-04611-f006:**
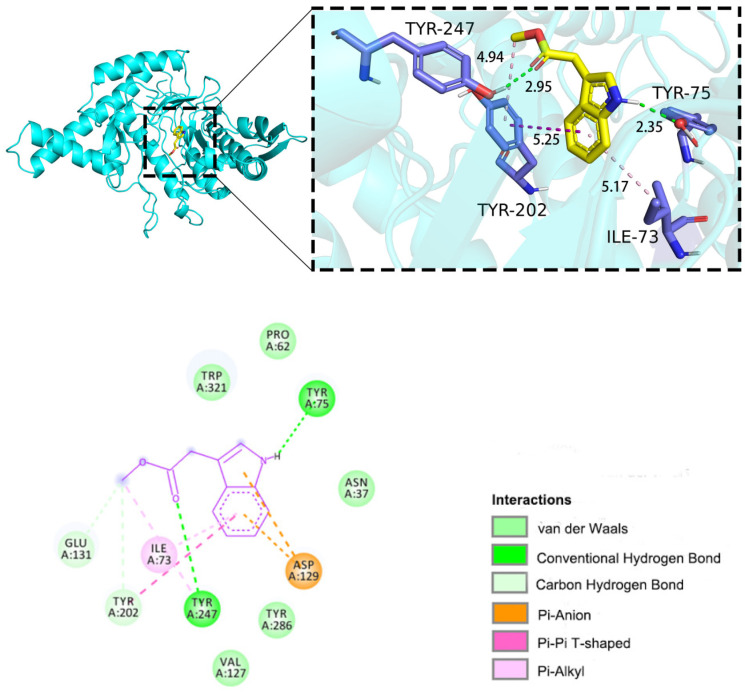
Molecular docking of methyl indole-3-acetate complex: 2D and 3D plots.

**Figure 7 ijms-25-04611-f007:**
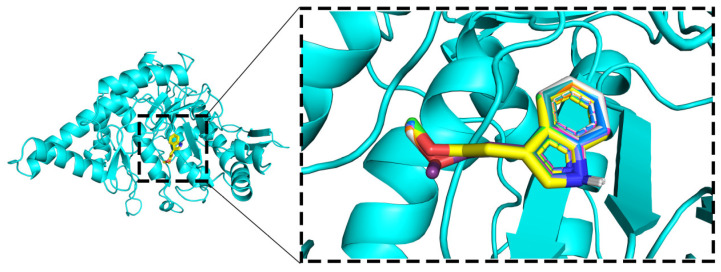
Visualization of ligand and protein interaction.

**Figure 8 ijms-25-04611-f008:**
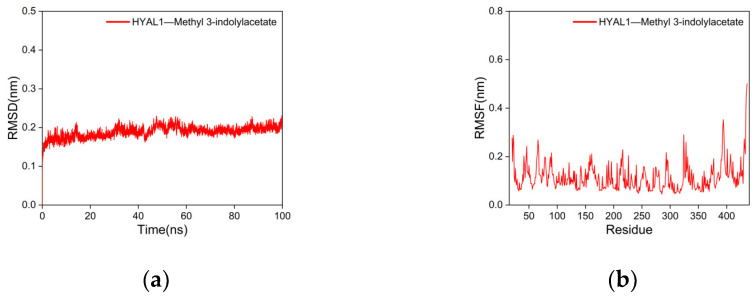
Results of molecular dynamics simulations. (**a**) RMSD curve; (**b**) RMSF curve.

**Figure 9 ijms-25-04611-f009:**
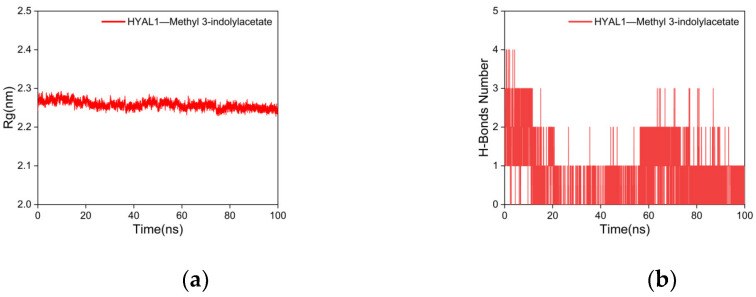
Results of molecular dynamics simulations. (**a**) Rg curve; (**b**) H-bond plot.

**Figure 10 ijms-25-04611-f010:**
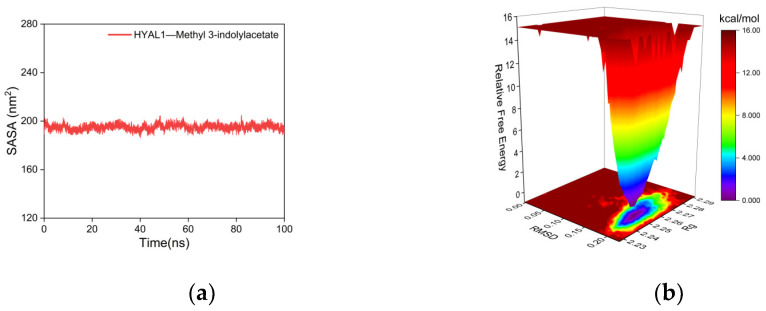
Results of molecular dynamics simulations. (**a**) SASA plot; (**b**) Gibbs FEL plot.

**Figure 11 ijms-25-04611-f011:**
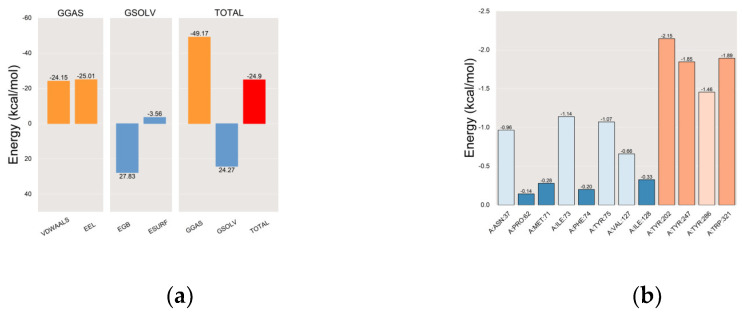
(**a**) MM-PBSA binding energy plot; (**b**) residue–energy plot.

**Figure 12 ijms-25-04611-f012:**
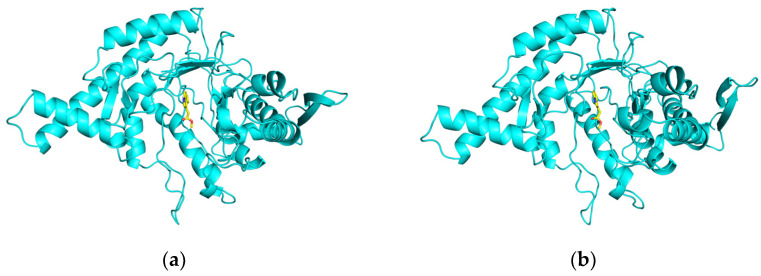
Binding mode of interaction in molecular dynamics simulations. (**a**) 25ns; (**b**) 50ns; (**c**) 75ns; (**d**) 100ns.

**Table 1 ijms-25-04611-t001:** Hyaluronidase-inhibitory activity of compounds isolated from the strain JNUCC 41.

**Compounds**	**Hyaluronidase-Inhibitory Activity** **IC_50_ (** **μM)**
Methyl indole-3-acetate	343.9
Dibutyl phthalate	-
Maculosin	-
N-Acetyl-L-tryptophan	-
EGCG	172.3

## Data Availability

Data are contained within the article and Appendix A.

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
