# Peer review of "Isolation, Characterization, Genome Annotation, and Evaluation of Hyaluronidase Inhibitory Activity in Secondary Metabolites of Brevibacillus sp. JNUCC 41: A Comprehensive Analysis through Molecular Docking and Molecular Dynamics Simulation"

_ijms, 2024, doi:10.3390/ijms25094611_

Round 1

Reviewer 1 Report

Comments and Suggestions for Authors

In the work of Xu, Y. et al. the Brevibacillus sp. JNUCC 41 was isolated, characterized, and subjected to comprehensive gene functional annotation. To further investigate the secondary metabolites governing biological functions, the strain was fermented, extracted, and isolated. Compounds exhibiting hyaluronidase inhibitory activity were screened, followed by an in-depth investigation of the inhibitory mechanism using molecular docking and molecular dynamics (MD) simulations. From my perspective, the article can be published with minor corrections.

-The authors provided a relevant and interesting discussion regarding the genes. As a suggestion to enhance comprehensiveness, the authors could include structures of orthologous proteins from the metagenomic atlas (https://esmatlas.com/resources/search_structure?query_id=aMoIhM9nmtWiMSJ4KRqJ8m7F01x8HgnQX6oIAQ) and add a discussion comparing them in the manuscript.

-In section 2.4, the authors showed that the molecule Methyl indole-3-acetate has advantages in ADMET compared to other molecules. However, the parameter "Plasma protein binding" is only 56.81%, indicating that the molecule will bind moderately to plasma transport proteins and will be poorly distributed throughout the body. No discussion or suggestions were presented regarding this. The authors should include this in the manuscript.

-In section 2.5.1, the authors wrote ".... A binding energy less than -5 kcal/mol signifies excellent binding affinity…". Without considering the sign of the binding energy, the term "less" could confuse readers. I suggest the authors replace "less" with "more negative."

-Figure 9 has poor quality; we can hardly visualize the interacting amino acids. The authors should improve the quality of the figure.

-The authors showed and discussed important interactions obtained in docking (lines 347 to 350) but did not present the binding distance. The authors should include this in the manuscript to demonstrate the strength of these interactions.

-In lines 373 and 374, the authors wrote "… The RMSD analysis results indicate that the complex formed between HYAL 1 and methyl indole-3-acetate exhibits good stability…". However, the simulations have a duration of only 100 ns, which is not sufficient to assess the stability of complexes. At least 500 ns would be preferable. The authors should modify their statements, removing the term "stability" and using "flexibility" instead.

-In lines 388 to 390, the authors stated "…. As shown in Figure 12a, the Rg curve of the complex formed between HYAL 1 and methyl indole-3-acetate fluctuates around 2.2-2.3 nm and remains stable throughout the simulation, without significant fluctuations…". If we analyze the Rg graph, we see a profile of a radius decreasing throughout the simulation. The authors should retract the claim of stability and discuss how the complex is becoming more compact over time.

-Along with the analysis of hydrogen bond numbers in Figure 12b, it would be interesting to add bond lengths to the manuscript to analyze the strength of these bonds.

-In the MMPBSA analyses, the authors could include a graph similar to Figure 14a in the manuscript, showing the contributions of each type of interaction (van der Waals, electrostatic, hydrogen bonding, and hydrophobic) to determine the predominant interaction. The discussion should be included in the manuscript.

Comments on the Quality of English Language

Minor editing of English language required

Author Response

< ijms-2930319>

< Isolation, Characterization, Genome Annotation, and Evaluation of Hyaluronidase Inhibitory Activity in Secondary Metabolites of Brevibacillus sp. JNUCC 41: A Comprehensive Analysis through Molecular Docking and Molecular Dynamics Simulation>

Dear Editor,

Thank you for your useful comments and suggestions on the language and structure of our manuscript. We have modified the manuscript accordingly, and detailed corrections are listed below point by point. We also dutifully rewrote the redundant sentences that our editor pointed out (in yellow).

Reviewer #1

  1. The authors provided a relevant and interesting discussion regarding the genes. As a suggestion to enhance comprehensiveness, the authors could include structures of orthologous proteins from the metagenomic atlas (https://esmatlas.com/resources/search_structure?query

_id=aMoIhM9nmtWiMSJ4KRqJ8m7F01x8HgnQX6oIAQ) and add a discussion comparing them in the manuscript.

→Thank you for your valuable suggestion. We have carefully considered your proposal regarding the inclusion of homologous protein structure comparisons. However, given the main focus of our current research and the complexity of data processing involved, we plan to explore this topic in future work and incorporate it as part of subsequent studies. We hope to have the opportunity to delve deeper into and implement this in future research endeavors.

  1. In section 2.4, the authors showed that the molecule Methyl indole-3-acetate has advantages in ADMET compared to other molecules. However, the parameter "Plasma protein binding" is only 56.81%, indicating that the molecule will bind moderately to plasma transport proteins and will be poorly distributed throughout the body. No discussion or suggestions were presented regarding this. The authors should include this in the manuscript.

→Thanks for your valuable suggestions, we have added a discussion about the predicted results of plasma protein binding. Due to the excessive number of charts in the manuscript, we have placed two tables of the ADMET section in the supplementary materials.

  1. In section 2.5.1, the authors wrote ".... A binding energy less than -5 kcal/mol signifies excellent binding affinity…". Without considering the sign of the binding energy, the term "less" could confuse readers. I suggest the authors replace "less" with "more negative."

→Thanks for your valuable suggestions, we have changed less than to more negative.

  1. Figure 9 has poor quality; we can hardly visualize the interacting amino acids. The authors should improve the quality of the figure.

→Thank you for your valuable suggestions. We have improved the image quality of Figure 9, and added the interaction distance.

  1. The authors showed and discussed important interactions obtained in docking (lines 347 to 350) but did not present the binding distance. The authors should include this in the manuscript to demonstrate the strength of these interactions.

→Thanks to your valuable suggestions, we added the interaction distance of the main force hydrogen bonding in the manuscript and added the other interaction distances in Figure 9.

  1. In lines 373 and 374, the authors wrote "… The RMSD analysis results indicate that the complex formed between HYAL 1 and methyl indole-3-acetate exhibits good stability…". However, the simulations have a duration of only 100 ns, which is not sufficient to assess the stability of complexes. At least 500 ns would be preferable. The authors should modify their statements, removing the term "stability" and using "flexibility" instead.

→Thanks for your valuable suggestions, we have replaced "stability' with "flexibility" in the manuscript.

  1. -In lines 388 to 390, the authors stated "…. As shown in Figure 12a, the Rg curve of the complex formed between HYAL 1 and methyl indole-3-acetate fluctuates around 2.2-2.3 nm and remains stable throughout the simulation, without significant fluctuations…". If we analyze the Rg graph, we see a profile of a radius decreasing throughout the simulation. The authors should retract the claim of stability and discuss how the complex is becoming more compact over time.

→Thank you for your valuable suggestions. We have removed stable and retained compact binding in the analysis of the Rg results section in the manuscript.

  1. Along with the analysis of hydrogen bond numbers in Figure 12b, it would be interesting to add bond lengths to the manuscript to analyze the strength of these bonds.

→Thanks for your valuable suggestions, we have added the analysis of hydrogen bond lengths in section 2.5.1 of the manuscript.

  1. In the MMPBSA analyses, the authors could include a graph similar to Figure 14a in the manuscript, showing the contributions of each type of interaction (van der Waals, electrostatic, hydrogen bonding, and hydrophobic) to determine the predominant interaction. The discussion should be included in the manuscript.

→Thanks for your valuable suggestions, we have added the discussion of MMPBSA analyzes to the manuscript.

Reviewer 2 Report

Comments and Suggestions for Authors

The article is very interesting, however, it requires improvement. The Materials and Methods chapter is placed in the wrong location. It should be placed after the Introduction chapter. This chapter lacks information on the specific methods used to assess pharmacokinetic parameters and drug-likeness. There is also a lack of methodology for evaluating the reliability of molecular docking simulations, including validation results. Can the findings of this study be extrapolated to other related enzymes or biological systems, and if so, what are the implications for future research? The Results and Discussion chapter is a simple comparison of the results of our own research with those of other authors. There is no attempt to explain the results.

The abstract requires revision as it is too general. It should include the key findings of the authors' research and the most important conclusions of the study.

The number of cited publications should be limited to those that make a significant contribution to the work. There is no need to cite a large number of publications containing the same content.

A significant deficiency in the presented work is the lack of a research hypothesis. The most challenging stage of research is formulating the research thesis and developing it into a research hypothesis, which must arise from a critical analysis of the scientific literature.

For the sake of conciseness and clarity, I believe certain revisions are necessary, such as shortening some passages and employing more concise formulations.

Author Response

< ijms-2930319>

< Isolation, Characterization, Genome Annotation, and Evaluation of Hyaluronidase Inhibitory Activity in Secondary Metabolites of Brevibacillus sp. JNUCC 41: A Comprehensive Analysis through Molecular Docking and Molecular Dynamics Simulation>

Dear Editor,

Thank you for your useful comments and suggestions on the language and structure of our manuscript. We have modified the manuscript accordingly, and detailed corrections are listed below point by point. We also dutifully rewrote the redundant sentences that our editor pointed out (in yellow).

Reviewer #2

  1. The article is very interesting, however, it requires improvement. The Materials and Methods chapter is placed in the wrong location. It should be placed after the Introduction chapter. This chapter lacks information on the specific methods used to assess pharmacokinetic parameters and drug-likeness. There is also a lack of methodology for evaluating the reliability of molecular docking simulations, including validation results. Can the findings of this study be extrapolated to other related enzymes or biological systems, and if so, what are the implications for future research? The Results and Discussion chapter is a simple comparison of the results of our own research with those of other authors. There is no attempt to explain the results.

→â‘ Thanks for your valuable suggestions, we have placed the Materials and Methods chapter after Introduction.

â‘¡Thank you for your valuable suggestions, we have added the prediction method for ADMET in section 2.9.

③Thank you for your valuable suggestions, the human hyaluronidase 1 (HYAL 1, PDB ID: 2PE4) has been widely studied and reported in docking simulations. however, without a co-crystallized inhibitor, it was not possible to perform corresponding comparative validations. Nevertheless, from the docking results, the binding of compounds to the protein exhibited good stability and strong interactions, and the role of key residues (Tyr-247 and Tyr-202) in molecular docking was consistently supported by MD simulations. We have revised the statement in the abstract to indicate that the crucial amino acid residues have also been confirmed in MD simulations,not " validation".

â‘£Thank you for your valuable suggestions. We conducted a comprehensive enzymatic activity screening on all isolated compounds, including tyrosinase, glucosidase, hyaluronidase, and elastase activities. Among them, only methyl indole-3-acetate exhibited previously unreported inhibitory activity against hyaluronidase, while the activities against other enzymes have been previously reported.

⑤Thank you for your valuable suggestions. We have modified the Results and Discussion chapter to focus on the analysis of the results.

  1. The abstract requires revision as it is too general. It should include the key findings of the authors' research and the most important conclusions of the study.

→Thank you for your valuable suggestions. We have modified the abstract section and highlighted the research hypotheses, methods, results, conclusions and significance.

  1. The number of cited publications should be limited to those that make a significant contribution to the work. There is no need to cite a large number of publications containing the same content.

 →Thank you for your valuable suggestions. We have deleted certain references to highlight important references.

  1. A significant deficiency in the presented work is the lack of a research hypothesis. The most challenging stage of research is formulating the research thesis and developing it into a research hypothesis, which must arise from a critical analysis of the scientific literature.

→Thank you for your valuable suggestions. After conducting essential analyses such as whole-genome sequencing and gene functional annotation on strains collected from Baengnokdam in Halla Mountain, we confirmed the strain's taxonomy and its active metabolism. Consequently, we proposed the experimental objective of studying the bioactivity of secondary metabolites from strain JNUCC41. We designed experimental methodologies including cultivation, isolation, characterization, enzyme assays, and docking simulations. Ultimately, we identified compounds exhibiting inhibition activity against hyaluronidase, indicating their potential applications in anti-inflammatory and skincare. Following your suggestions, we have modified the abstract and conclusion sections, emphasizing the research hypothesis, objectives, results, and application potentials.

  1. For the sake of conciseness and clarity, I believe certain revisions are necessary, such as shortening some passages and employing more concise formulations.

→Thank you for your valuable suggestions. We have simplified the entire manuscript to a certain extent, emphasizing conciseness and clarity.

Reviewer 3 Report

Comments and Suggestions for Authors

This manuscript is important and interesting but there is a lack of presentation. It is too long (15 figures, 3 tables and 109 references). The size could be reduced by switching some figures and less important data to Supplementary info. 

The conclusion is also necessary to revise without using any references and reduce the words as much as possible. 

The number of references should be reduced with updated relevant findings from the existing public literature. 

Author Response

< ijms-2930319>

< Isolation, Characterization, Genome Annotation, and Evaluation of Hyaluronidase Inhibitory Activity in Secondary Metabolites of Brevibacillus sp. JNUCC 41: A Comprehensive Analysis through Molecular Docking and Molecular Dynamics Simulation>

Dear Editor,

Thank you for your useful comments and suggestions on the language and structure of our manuscript. We have modified the manuscript accordingly, and detailed corrections are listed below point by point. We also dutifully rewrote the redundant sentences that our editor pointed out (in yellow).

  1. This manuscript is important and interesting but there is a lack of presentation. It is too long (15 figures, 3 tables and 109 references). The size could be reduced by switching some figures and less important data to Supplementary info. 

→Thank you for your valuable suggestions. We have deleted certain references to highlight important references, and placed two tables of the ADMET section in the supplementary materials.

  1. The conclusion is also necessary to revise without using any references and reduce the words as much as possible. 

→Thank you for your valuable suggestions. We have revised the conclusion.

  1. The number of references should be reduced with updated relevant findings from the existing public literature. 

→Thank you for your valuable suggestions. We have deleted certain references to highlight important references.

Round 2

Reviewer 2 Report

Comments and Suggestions for Authors

 Accept in present 

Author Response

Reviewer 2 has no comments, so you upload only the revised version of the paper.

Reviewer 3 Report

Comments and Suggestions for Authors

The authors addressed most of my concerns and modified the manuscript. However, there are still some issues and those could be fixed. Especially, the size of the manuscript. Although the authors switched two tables, I would suggest switching several less important images to the supplementary info. 

Author Response

Reviewer #3

The authors addressed most of my concerns and modified the manuscript. However, there are still some issues and those could be fixed. Especially, the size of the manuscript. Although the authors switched two tables, I would suggest switching several less important images to the supplementary info. 

→Thanks for your valuable suggestions.. We have deleted three images (sections 3.1.1, 3.1.4, and 3.1.5) and placed the predicted data (section 3.1.1) in the Supplementary materials.